# Ultramicronized *N*-Palmitoylethanolamine Regulates Mast Cell-Astrocyte Crosstalk: A New Potential Mechanism Underlying the Inhibition of Morphine Tolerance

**DOI:** 10.3390/biom13020233

**Published:** 2023-01-25

**Authors:** Alessandra Toti, Laura Micheli, Elena Lucarini, Valentina Ferrara, Clara Ciampi, Francesco Margiotta, Paola Failli, Chiara Gomiero, Marco Pallecchi, Gianluca Bartolucci, Carla Ghelardini, Lorenzo Di Cesare Mannelli

**Affiliations:** 1Department of Neuroscience, Psychology, Drug Research and Child Health—NEUROFARBA—Pharmacology and Toxicology Section, University of Florence, Viale Pieraccini 6, 50139 Florence, Italy; 2Epitech Group SpA, Via Luigi Einaudi 13, 35030 Padua, Italy; 3Department of Chemistry, University of Florence, Via Ugo Schiff 6, 50019 Florence, Italy

**Keywords:** morphine, tolerance, *N*-palmitoylethanolamine, mast cell, glial cells, cross-talk

## Abstract

Persistent pain can be managed with opioids, but their use is limited by the onset of tolerance. Ultramicronized *N*-palmitoylethanolamine (PEA) in vivo delays morphine tolerance with mechanisms that are still unclear. Since glial cells are involved in opioid tolerance and mast cells (MCs) are pivotal targets of PEA, we hypothesized that a potential mechanism by which PEA delays opioid tolerance might depend on the control of the crosstalk between these cells. Morphine treatment (30 μM, 30 min) significantly increased MC degranulation of RBL-2H3 cells, which was prevented by pre-treatment with PEA (100 μM, 18 h), as evaluated by β-hexosaminidase assay and histamine quantification. The impact of RBL-2H3 secretome on glial cells was studied. Six-hour incubation of astrocytes with control RBL-2H3-conditioned medium, and even more so co-incubation with morphine, enhanced CCL2, IL-1β, IL-6, Serpina3n, EAAT2 and GFAP mRNA levels. The response was significantly prevented by the secretome from PEA pre-treated RBL-2H3, except for GFAP, which was further upregulated, suggesting a selective modulation of glial signaling. In conclusion, ultramicronized PEA down-modulated both morphine-induced MC degranulation and the expression of inflammatory and pain-related genes from astrocytes challenged with RBL-2H3 medium, suggesting that PEA may delay morphine tolerance, regulating MC-astrocyte crosstalk.

## 1. Introduction

Pain management is still one of the most significant problems that affect patient quality of life [1]. Pain can result from injury [2], surgery [3] or disease, such as type 1 and 2 diabetes, osteoarthritis as well as cancer [4,5,6] and it can also arise as a side effect of anti-cancer chemotherapies [7]. Therapies for pain relief depend on the intensity or persistence of pain; for mild pain acetaminophen, non-steroidal anti-inflammatory drugs (NSAIDs) such as aspirin, ibuprofen and naproxen are recommended, while in the case of severe pain, opioids such as morphine are the first choice [8]. 

The analgesic effect of morphine is mainly exerted by μ-receptor [9], although morphine also interacts, albeit weakly, with the κ and δ receptors [10]. Morphine receptors are distributed in the peripheral and central nervous system (PNS and CNS), especially in the periaqueductal gray matter and posterior horns of the spinal cord, but they are also present in different peripheral tissues and organs, being responsible for several side effects, including vomiting, nausea, constipation, pruritus, and respiratory depression. These effects typically arise within the first weeks of treatment and are likely to become more severe with time, when cognitive failure, tolerance and addiction may develop [11]. 

In particular, tolerance is defined as the necessity to increase the dose of morphine in order to obtain a good analgesic effect after repeated treatment and can be managed with opioid rotation, as long as the side effects do not lead to treatment withdrawal [12]. In recent decades, morphine tolerance has been attributed to the desensitization/internalization of μ-receptors and the involvement of glial cell overactivation has been recently suggested [13,14]. 

Glial cells (i.e., astrocytes and microglia together with oligodendrocytes and Schwann cells), not only exert supportive functions, but are increasingly considered to play a pivotal role in the transmission of pain and regulation of numerous pathophysiological processes [15,16,17]. In rats daily administered with morphine 10 mg/kg i.p. a significant increase in the number of glial cells was observed in the dorsal horn of the spinal cord, once tolerance has developed (i.e., after 6 days) [18]. Accordingly, it has been recently demonstrated that glial selective inhibitors as fluorocitrate, minocycline, propentofylline, and pentoxifylline delay morphine tolerance [19,20,21,22]. 

*N*-palmitoylethanolamine (PEA) is a glia modulator and a pro-homeostatic lipid mediator, belonging to the family of fatty acid ethanolamines, which are produced on demand from the lipid bilayer [23]. Recent studies have demonstrated the efficacy of PEA against persistent pain [24] and neurological disorders [25] with an optimal safety profile in humans [26]. Increasing information is currently available on the pleiotropic mechanism of action of PEA, which behaves as a cannabinomimetic compound, stimulating the effects of endocannabinoids through the so-called entourage effect [27]. Moreover, PEA is an agonist of the peroxisome proliferator-activated receptor type α (PPAR-α), a nuclear transcription factor able to inhibit both pro-inflammatory cytokine release [28] and the activity of enzymes such as cyclooxygenase [29]. In a model of chronic constriction injury, we have previously demonstrated that pain-relieving effects of PEA depend, at least in part, on a PPAR-α-mediated mechanism [30]. 

Notably, treatment with a bioavailable PEA formulation (i.e., ultramicronized PEA) was shown to delay the onset of morphine tolerance and prevent the increase of glial cells number in the dorsal horn of the spinal cord [18]. The mechanisms sustaining this effect of PEA are still unclear. It is known that morphine activates mast cells (MCs) in vivo and enhances histamine release [31,32,33], while PEA down-modulates MC degranulation [34] and astrocyte activity [35]. 

Based on this background, the aim of the present study was to investigate if a crosstalk exists between astrocytes and MCs during morphine treatment and if PEA can regulate it. Despite the well-known limitations of 2D cell cultures, the present study might provide novel insights in the mechanism(s) underlying PEA-induced delay of morphine tolerance and non-neuronal cell(s) mainly involved in. The results could thus improve our understanding on the management of opioid tolerance and related neuroinflammatory disorders.

## 2. Materials and Methods

### 2.1. Materials

Ultramicronized PEA (99.9% powder particle size <6 μm) was kindly provided by Epitech Group (Saccolongo, Italy) and for simplicity will be referred to as PEA hereafter. PEA was dissolved in dimethyl sulfoxide (DMSO, Merck, Milan, Italy) at 10 mM, and 100 µM was used with a final concentration of 1% DMSO. Morphine hydrochloride was purchased from S.A.L.A.R.S. (Como, Italy) and dissolved in sterile water.

### 2.2. Animals

The Sprague-Dawley rats were purchased from the Envigo company (Varese, Italy) around the seventeenth day of pregnancy. They were housed in a 26 cm × 41 cm cage at the Laboratory Animal Housing Center (Ce.SAL) of the University of Florence, fed with a standard diet and water ad libitum and kept at 23 ± 1 °C with a light cycle/12 h dark, with light at 7 am. The rats were kept alive until delivery and the pups were sacrificed between day 1 and day 3 of life to isolate a primary culture of rat cortical astrocytes. All handling of animals were carried out in accordance with the guidelines of the European Community for the care of animals (DL 116/92, application of the directives of the Council of the European Community of 24 November 1986; 86/609/EEC). The ethics policy of the University of Florence complies with the decree Guide for the Care and Use of Laboratory Animals of the US National Institutes of Health (NIH Publication number 85-23, revised 1996; University of Florence assurance number: A5278-01). Formal approval to conduct the experiments described was obtained from the Italian Ministry of Health (No. 333/2020-PR) and from the Animal Subjects Review Board of the University of Florence. All efforts were made to minimize animal suffering and reduce the number of animals used. 

### 2.3. Cell Culture

The rat basophilic leukemia cells (RBL)-2H3, purchased from American Type Culture Collection (ATCC, Manassas, VA, USA), were cultured in Eagle’s Minimum Essential Medium (EMEM)-High Glucose growth medium (ATCC, Manassas, VA, USA), containing L-glutamine (2 mM), supplemented with 15% Heat Inactivated Fetal Bovine Serum (FBS, Euroclone, Milan, Italy), penicillin (100 U/mL) and streptomycin (100 μg/mL; Merck, Milan, Italy). The RBL-2H3 cell line was used as a commonly accepted model of MCs [36].

Primary cultures of cortical astrocytes were obtained according to the method described by McCarthy and de Vellis [37]. Briefly, the cerebral cortex of new-born Sprague-Dawley rats (Envigo, Varese, Italy) post-natal day 1–3 was dissociated in Hanks balanced salt solution containing 0.5% trypsin, 0.2% EDTA and 1% DNase (Merck, Milan, Italy) for 30 min at 37 °C. The suspension was mechanically homogenized and filtered with 70 μm filters. Cells were plated in Dulbecco’s Modified Eagle’s Medium (DMEM)-High Glucose (Merck, Milan, Italy) supplemented with 20% FBS (Euroclone, Milan, Italy), L-glutamine (2 mM), penicillin (100 U/mL) and streptomycin (100 μg/mL; Merck, Milan, Italy). Confluent primary glial cultures were used to isolate astrocytes, removing microglia and oligodendrocytes by shaking. After 21 days of culture, astrocytes were plated according to experimental requirements. Cells were maintained in incubator, at 37 °C, and in a humidified atmosphere of 5% CO_2_.

### 2.4. Cell Viability Assay

The RBL-2H3 cells and astrocytes were plated at the density of 4 × 10^3^ and 5 × 10^3^ cells/well respectively in a 96-well plate. After the appropriate treatments at 24 and 48 h, the medium was removed and the cells incubated with a 1 mg/mL solution of Thiazolyl Blue Tetrazolium Bromide (MTT; Merck, Milan, Italy), at 37 °C, for 1 h and a half. The dissolution of the crystals was carried out with 200 µL of DMSO per well, and the absorbance was read at 590 nm using the EnSight Multimode Plate Reader (Perkin Elmer) [38]. 

### 2.5. Cell Count

The RBL-2H3 cells were plated in 6-well plate, at a density of 4 × 10^5^ cells/well, sensitized with anti-DNP IgE (50 ng/mL) and concurrently treated with PEA 100 μM. After 24 and 48 h, cells were detached and counted using a Burker’s chamber. A mean of two counts for condition with 4 replicates were analyzed [39].

### 2.6. β-hexosaminidase Assay

The RBL-2H3 cells were plated in a 24-well plate, at a density of 5 × 10^4^ cells/well, and the next day cells were sensitized with anti-dinitrophenyl (DNP)-IgE (50 ng/mL) at 37 °C overnight. After 18 h, two washes with modified Tyrode’s (MT) buffer (20 mM HEPES, 135 mM NaCl, 5 mM KCl, 1.8 mM CaCl_2_, 1 mM MgCl_2_, 5.6 mM glucose, and 0.05% BSA, pH 7.4) were performed and then cells were treated in 250 μL of MT buffer containing morphine 30 μM, PEA 10–100 μM or morphine 30 μM + PEA 100 μM for 30 min at 37 °C. Cells were then stimulated with DNP-BSA (625 ng/mL) for 5, 10, 15 and 30 min at 37 °C. After stimulation, the supernatant was collected and the cells lysed with 250 μL of MT buffer containing 0.1% Triton X-100, and then sonicated for 30 s on ice. Both supernatants and cell lysates were transferred to a new 96-well plate, 50 μL per sample, and incubated for 1 h at 37 °C with 50 μL/well of 1 mM p-nitrophenyl-N-acetyl-d-glucosaminide (NAG; Merck, Milan, Italy) dissolved in 0.1 M citrate buffer (pH 4.5). The enzymatic reaction was then stopped by adding 100 μL/well of 0.1 M carbonate buffer (pH 10.0). The absorbance (A) was read at 405 nm using EnSight Multimode Plate Reader (Perkin Elmer). The percentage of β-hexosaminidase released was calculated as follows: 100 × [(A supern − A white)/(A supern − A white) + (A pellet − A white pellet)] [40].

### 2.7. Histamine Assay 

The amount of histamine in RBL-2H3 media was measured by isotopic dilution (ID) bidimentional high performance liquid chromatography (2D-HPLC) with tandem mass spectrometry (MS/MS) method. The instrument employed was a Varian (Palo Alto, CA, USA) triple quadrupole 1200 L system (able to perform MS/MS experiments) coupled with three Prostar 210 pumps (each one can manage one solvent line), a Prostar 410 autosampler and an electro-spray ionization (ESI) source. All the raw data collected were processed by Varian Workstation (version 6.8) software. The system was equipped with two 6-port valves. The following columns were used: (1) SeQuant^®^ ZIC-HILIC 20 × 2.1 mm as loading column and (2) SeQuant^®^ ZIC-HILIC 50 × 2.1 mm as analytic column, both with a particle size of 3.5 µm. The elution was performed with the following solvents: (1) solvent A, mQ water:acetonitrile 9:1 solution added with 5 mM formic acid and 15 mM ammonium formate; (2) solvent B, mQ water:acetonitrile 1:9 solution added with 15 mM formic acid and 5 mM ammonium formate; and (3) solvent C (used for sample loading), mQ water:acetonitrile 1:9 solution added with 17.5 mM formic acid 2.5 mM ammonium formate. The loading time was carried out in isocratic elution with the solvent C for 2 min at flow of 0.5 mL min^−1^. It was demonstrated that in 2 min the analyte is completely retained by the loading column. After 2 min the valve changed position and the gradient of A and solvents started in counter-flow from the loading to the analytic columns. The gradient started from 90% to 20% of solvent B in 7 min and held for 3 min, then it returned at 90% of solvent B in 0.1 min and restored the initial condition for 13 min, for a total run time of 25 min [41]. More details on the method are reported in Appendix A.

### 2.8. Collection of Mast Cell-Conditioned Medium

RBL-2H3 media were collected after appropriate treatment and centrifuged two times for 5 min at 200× *g* to eliminate floating cells and debris before treating astrocytes.

### 2.9. Timeline of Astrocytes Treatment with Mast Cell-Conditioned Medium

Primary astrocytes were plated in 6-well plate at a density of 2.5 × 10^4^ cells/well and subsequently treated for 48 h with 30 μM morphine in DMEM with 1% FBS. In parallel, RBL-2H3 were plated in 6-well plate at a density of 4 × 10^5^ cells/well and pre-sensitized the next day with anti DNP-IgE 50 ng/mL ± PEA 100 μM. 

The next day, after two washes in phosphate buffer solution (PBS) to eliminate DNP-IgE and PEA, the RBL-2H3 were incubated in DMEM with 1% heat inactivate FBS and 0.05% BSA and then stimulated for 24 h with DNP-BSA 625 ng/mL. Conditioned media of RBL-2H3 were collected as previously described. At the end of 48 h of morphine treatment, astrocytes were incubated for 6 h with media collected from control and PEA pre-treated RBL-2H3 (see Figure 1).

### 2.10. Protein Extraction, Electrophoresis, and Western Blot

Cells were lysed with 50 mM Tris-HCl pH 8.0, 150 mM NaCl, 1 mM EDTA, 0.5% Triton X-100 and complete protease inhibitors (Roche, Milan, Italy). 

The lysates were collected, sonicated for 3 min in ice, vortexed for 1 min and finally centrifuged at 12,000× *g* for 10 min, at 4 °C, and the supernatant collected. Total protein concentrations were quantified by bicinchoninic acid test (Merck, Milan, Italy). An amount of 40 μg of proteins was resolved with precast polyacrylamide gel (BOLT 4–12% Bis-Tris Plus gel; Thermo Fisher Scientific, Monza, Italy) before electrophoretic transfer to nitrocellulose membranes (Bio-Rad, Milan, Italy). The membranes were blocked with 5% Blotto, non-fat dry milk (Santa Cruz Biotechnology, Dallas, TX, USA) dissolved in PBS-0.1% Tween (PBST; Merck, Milan, Italy) and then probed overnight at 4 °C with primary antibodies specific for mouse MOR-1 (1:1000; cod. sc-515933, Santa Cruz Biotechnology, Dallas, TX, USA) and mouse GAPDH (1:5000; cod. sc-32233, Santa Cruz Biotechnology, Dallas, TX, USA). The membranes were then incubated for 1 h in PBST containing the anti-mouse secondary antibody HRP conjugated (1:5000; Santa Cruz Biotechnology, Dallas, TX, USA). ECL (Enhanced Chemiluminescence Pierce, Rockford, IL, USA) was used to visualize peroxidase-coated bands. Densitometric analysis was performed using the ImageJ analysis software (ImageJ; NIH, Bethesda, MD, USA). Normalization with housekeeping gene GAPDH content was performed [42]. 

### 2.11. Immunofluorescence

The RBL-2H3 cells and astrocytes were plated at the density of 4 × 10^5^ and 5 × 10^4^ cells/well respectively on 22 × 22 mm cover slides placed in a 6-well plate. After the appropriate treatments, the cells were fixed with 4% paraformaldehyde for 15 min at room temperature. Permeabilization was performed by incubation for 1 h with PBS 1X-0.3% Triton and blocking phase with PBS 1X-0.3% Triton+0.5% BSA followed by overnight incubation at 4 °C with rabbit anti-mast cell tryptase (1:200, GeneTex, TX, USA) or rabbit anti-glial fibrillary acidic protein (GFAP; 1:200, cod. Z0334, Dako Agilent, Santa Clara, CA, USA) diluted in the blocking solution. The next day, after appropriate washing in PBS, the slides were incubated for 2 h with the Alexa Fluor ^®^ 568 goat anti-rabbit conjugated secondary antibody (1:500, Life Milan, Italy) at room temperature, diluted in the blocking solution. Finally, nuclei were labeled with 4′,6-diamidine-2-phenylindole (DAPI; Life Technologies, Milan, Italy) for 5 min, diluted in PBS to a final concentration of 500 ng/mL. After washing in double distilled water, the slides were placed on a specimen slide using the Fluoromount^TM^ aqueous mounting medium (Life Technologies, Milan, Italy). The images were acquired using a Leica DM6000 B motorized fluorescence microscope equipped with a DFC350FX camera (Leica, Mannheim, Germany). The intensity was calculated using imageJ and analyzing 10 fields per slide acquired randomly with a 20× objective [42].

### 2.12. Real Time Polymerase Chain Reaction (RT-PCR)

Total RNA was isolated from cells using NucleoSpin^®^ RNA (Macherey-Nagel, Düren, Germany) following manufacturer’s instruction. An amount of 500 nanograms of RNA was retrotranscribed using iScript cDNA Synthesis kit (Bio-Rad, Milan, Italy). RT-PCR was performed using SsoAdvanced Universal SYBR^®^ Green Supermix (Bio-Rad, Milan, Italy) following the thermal profile suggested by the kit.

The following primers were used for detection: 

rSerpina3n: forward 5′-CTTTCTGCAGTATGTGGGAATCACTTGG-3′ and reverse 5′-GGCTGCATTGCTCTAAGTAGGAGTGC-3′ (Invitrogen); 

rCCL2: forward 5′-TCTTCCTCCACCACTATGCAGGTCTC-3′ and reverse 5′-TCTTTGGGACACCTGCTGCTGGTG-3′ (Invitrogen); 

rGFAP forward 5′-CTGACACACGTTGTGTTCAAGCAGCC-3′ and reverse 5′-CTGAAGGTTAGCAGAGGTGACAAGGG-3′ (Invitrogen).

Validated primers to detect rIL-6 (qRnoCID0053166), rIL-1β (qRnoCID0004680), rEAAT2 (qRnoCED0005967), rTNFα (qRnoCED0009117), and rβ2microglobulin as housekeeping (qRnoCED0056999), were purchased from Bio-Rad (Bio-Rad, Milan, Italy). The differential expression of the transcripts was normalized on the housekeeping gene.

### 2.13. Measurement of Intracellular Ca^2+^

Dynamic intracellular cytosolic Ca^2+^ ([Ca^2+^]i) was evaluated in astrocytes plated on round glass coverslips (diameter 25 mm) and after the appropriate treatments they were labeled with 4 µM Fluo-4AM (Life technologies, Milan, Italy) for 45 min, at 37 °C. Cells were washed and kept in Hepes buffer (NaCl 140 mM, NaHCO_3_ 12 mM, Hepes 10 mM, KCl 2.9 mM, NaH_2_PO_4_ 0.5 mM, CaCl_2_ 1.5 mM, glucose 10 mM, MgCl_2_ 1.2 mM, pH = 7.4). The coverslips were mounted in a perfusion chamber and placed on the stage of an inverted reflected light fluorescence microscope (Zeiss Axio Vert. A1 FL-LED) equipped with fluorescence excitation (475 nm) with LED. Cells were incubated for at least 5 min in the control solution and then stimulated with 100 μM ATP. The dynamics of calcium was recorded for at least 3 min after administration of the agonist. The fluorescence of Fluo-4AM was recorded with a Tucsen Dhyana 400D CMOS camera (Tucsen Photonics Co., Ltd., Fuzhou, China) with a resolution of 1024 × 1020 pixels^2^. Ca^2+^ dynamics was measured by single cell imaging analysis at 35 °C. The images were recorded using the Dhyana SamplePro software, recording 13 frames per second and dynamically analysed with the open-source community software for bioimaging Icy (Pasteur Institute, Paris, France). A signal-to-noise ratio of at least 5 arbitrary units of fluorescence (A.U) was considered as an agonist-induced increase in Ca^2+^. The following parameters were evaluated: the ratio between the agonist-induced maximum change in fluorescence and the basal fluorescence (ΔF/F, measured as A.U.), the time to reach the maximum calcium value and the area under the fluorescence calibrated curve baseline (AUC/F, evaluated as A.U.). The experiments were repeated in 3 different cell preparations, analysing at least 6 cells per optical field (using a 40× magnification objective) [43].

### 2.14. Statistical Analysis

Results were expressed as mean ± S.E.M. and analysis of variance (ANOVA) was performed.

A Bonferroni significant difference procedure was used as post hoc comparison. Data were analysed using the “Origin 8.1” software (OriginLab, Northampton, MA, USA). Differences among groups were considered significant at values of *p* < 0.05. * *p*-value < 0.05; ** *p*-value < 0.01; *** *p*-value < 0.001.

## 3. Results

### 3.1. Morphine and PEA, Used Solely or in Combination, Do Not Affect Mast Cell Viability

We first verified by MTT test that none of the investigated concentrations of morphine (0.3–300 µM) was toxic on RBL-2H3 cells at either 24 or 48 h treatment (Figure 1a). The 30 µM concentration was therefore chosen since it was observed to induce a consistent degranulation without any toxic effects. Then, increasing concentrations of PEA (10 nM–100 µM) together with 30 µM morphine were tested on RBL-2H3 cells by MMT. Figure 1b,c show that PEA at the concentration range 10 nM–10 µM, alone or in combination with 30 µM morphine, does not affect cell viability in a significant manner, either at 24 or 48 h. Although a decrease in RBL-2H3 cell viability was observed with PEA 50 µM and 100 µM, with the effect being worsened by morphine, PEA 100 µM (24 h treatment) did not affect the cell count, suggesting that it was not toxic at the higher concentration (Figure 1d). A decrease in the number of cells was observed at 48 h, probably due to an impaired proliferation more than a toxic effect of PEA (Figure 1d). For this reason, 24 h treatment was selected for further experiments. The total protein content at 24 h in control and PEA 100 µM-treated RBL-2H3 cells confirmed the findings from the cell count assay (Figure 1e).

### 3.2. RBL-2H3 Cells Express the µ-Receptor and Morphine Augments Mast Cell Degranulation 

Since opioid treatment is known to activate MCs [33], but no information is available on the molecular mechanism(s), we first evaluated the expression of morphine receptors on RBL-2H3 cells. Western blot analysis of MOR-1 demonstrated, for the first time, that RBL-2H3 cells express morphine µ-receptor (Figure 2a and Appendix A). Given the consistent basal degranulation, the RBL-2H3 cell activation protocol had first to be optimized (Appendix A) before investigating morphine-induced degranulation. Similar to what reported by other groups [44,45], RBL-2H3 cells were pre-sensitized with anti-DNP IgE (50 ng/mL) for 18 h. The next day, cells were incubated with morphine (0.3–300 μM) for 30 min in a buffer containing 0.05% BSA and then stimulated with DNP-BSA (625 ng/mL) for 10 min [46,47]. In these conditions, a controlled granule release was achieved. As reported in Figure 2b, morphine induced a concentration dependent decrease of tryptase intensity, suggesting an increased tryptase release and thus degranulation. To better investigate morphine-induced degranulation, the release of β-hexosaminidase was also studied and the percentage of β-hexosaminidase in the medium was compared to that inside the cells. A significant 15% increase in the release of β-hexosaminidase was observed in sensitized RBL-2H3 cells challenged with DNP-BSA for 5, 10, 15 and 30 min and treated with 30 μM morphine for 30 min (Figure 2c). Lengthening morphine incubation time up to 24 h did not generate any statistically significant increase (Figure 2d), probably due to the very short time needed by morphine to induce maximum RBL-2H3 cell degranulation.

### 3.3. PEA Counteracts Morphine-Augmented Mast Cell Degranulation

Since a previous in vivo study in rats demonstrated that one-week pretreatment with PEA enhanced the analgesic effect of morphine, thus exerting an opioid sparing effect [48], we choose to pre-treat RBL-2H3 cells with PEA during anti DNP-IgE sensitization (18 h) and then incubate cells with PEA for 30 min the following day. The incubation time and doses were based on preliminary experiments (Appendix A). Briefly, the β-hexosaminidase assay revealed that PEA 10 µM was effective in reducing RBL-2H3 cell degranulation up to 40%, while PEA at 100 µM achieved 80% reduction (Appendix A). The ability of 100 µM PEA to counteract morphine-induced RBL-2H3 cell degranulation was analysed by β-hexosaminidase assay. RBL-2H3 cells were challenged with DNP-BSA for 5, 10, 15 and 30 min. Interestingly, the effect of PEA was found to be statistically significant not only compared to morphine, but also with regard to the DNP-BSA-challenged RBL-2H3 cells (Figure 3a). Moreover, the second pre-treatment period with PEA (100 µM for 30 min, after the first 18 h pre-treatment) appeared not to be necessary, since the inhibitory activity was maintained after a single pre-treatment period both in the absence and presence of morphine 30 μM (Figure 3a). To confirm these data, the level of histamine released in the medium was also quantified by HPLC with the same experimental setting. The assay allowed to identify a slight, albeit non-significant, increase in histamine concentrations in 30 μM morphine-treated RBL-2H3 cells compared to controls. Conversely, pre-treatment with PEA 100 μM significantly decreased histamine release, both in the absence and presence of morphine 30 μM (Figure 3b). 

### 3.4. PEA Down-Modulation of Mast Cell Degranulation Is Long-Lasting

To investigate whether down-modulation of RBL-2H3 cells by PEA (100 μM) was long-lasting, we challenged cells with DNP-BSA for 30 min, 2, 6 and 24 h the day after anti DNP-IgE sensitization and measured the content of histamine in the medium at every timepoint. After each challenge, PEA-treated RBL-2H3 cells released a significantly lower amount of histamine compared to controls. As shown in Figure 4a, histamine concentration (µM) in treated and control cells was respectively 2.55 ± 0.1 vs. 10.2 ± 0.13 at 30 min, 6.86 ± 0.71 vs. 15.07 ± 0.5 at 2 h, 24.57 ± 0.1 vs. 39.03 ± 0.7 at 6 h and finally 89.98 ± 0.9 vs. 126.92 ± 0.9 at 24 h. PEA 100 μM significantly increased also tryptase immunofluorescence, thus indicating a decreased degranulation (Figure 4b). Since MC activation is known to induce not only the release of pre-formed mediators (e.g., histamine) but also the release of de-novo synthesized cytokines and chemokines, the effect of PEA on newly synthesized mediators was also investigated. Using RT-PCR, we found that 18 h pre-treatment with PEA 100 μM before 6 h DNP-BSA challenge, efficiently counteracted the increase in the expression of the CCL2 (0.1 ± 0.002 vs. 1 ± 0.02) and TNF-α (0.63 ± 0.015 vs. 1 ± 0.02) genes, compared to controls (Figure 4c).

### 3.5. PEA Down-Modulates Mast Cell-Induced Astrocyte Expression of Genes Involved in Inflammation and Pain 

First, the effect of morphine and PEA on primary cortical astrocyte viability was evaluated as better detailed in the Appendix A. Briefly, cell viability was not affected by increasing concentrations of morphine (1–30 μM) while PEA (10–30 μM) affected astrocyte viability, either used alone or in combination with morphine (Appendix A). Second, the presence and functional expression of μ-receptors was studied, using western blot and calcium influx respectively, confirming that primary cortical astrocytes express a functional μ-receptor (Appendix A). The effect of RBL-2H3 cell mediators (with or without PEA treatment) on astrocyte viability and activity (with or without morphine treatment) was then studied by incubating the primary cortical astrocytes with RBL-2H3 cell medium. Briefly, we showed that RBL-2H3 cell-conditioned medium does not affect astrocytes viability (Figure 5a). Furthermore, by using RT-PCR we showed that morphine treatment does not alter the astrocyte expression of any of the investigated genes, i.e., GFAP, EAAT2, Serpina3n, IL-1β, IL-6, CCL2. On the contrary, control astrocytes incubated with control RBL-2H3 cell medium showed a significant increase in the expression of all the genes but EAAT2 (Figure 5b,c). Cell medium from PEA pre-treated RBL-2H3 cells significantly reversed the increased expression for all the investigated genes, except GFAP that was furtherly increased (Figure 5b,c). The morphine-treated astrocytes incubated with control RBL-2H3 cell media showed a significant increase in the expression of all the genes examined, including EAAT2, with values exceeding those in control astrocytes treated with the same medium. The incubation with cell medium from PEA pre-treated RBL-2H3 cells significantly reversed the increased expression of all genes, except for GFAP (Figure 5b,c). In order to better understand the apparently contradictory findings on GFAP gene expression, immunofluorescence analysis of GFAP was performed. In contrast to RT-PCR findings, increased GFAP staining was observed in 30 μM morphine-treated astrocytes. Similar findings were also seen in control (and even more so in morphine-treated astrocytes) incubated with control RBL-2H3 cell medium. In line with RT-PCR findings, the cell medium from PEA pre-treated RBL-2H3 cells further enhanced GFAP staining compared to control RBL-2H3 cell medium, both in control and morphine-treated astrocytes (Figure 6). Summarizing, morphine alone did not change the gene expression profile of astrocytes, while increasing GFAP immunoreactivity. On the contrary, a significant increase of the investigated genes was evident if mediators from RBL-2H3 cells were added. Finally, pre-treatment of RBL-2H3 cells with PEA significantly down-modulated the expression of astrocyte genes involved in neuropathic pain and inflammation. 

## 4. Discussion

The management of persistent pain is a major challenge for clinicians as well as patients. Opioids are the treatment of choice, but their use is limited by the occurrence of side effects such as tolerance. We have recently shown that ultramicronized PEA effectively delays tolerance to morphine, tramadol and oxycodone [18,41]. Moreover, ultramicronized PEA was shown to enhance the analgesic effect of opioids, thus limiting their dosage and side effects, both in naïve and neuropathic animals [48,49]. However, the mechanisms underlying the ability of PEA to reduce opioid tolerance are still unclear. 

In light of the evidence that (i) PEA down-modulates MC degranulation [50] and (ii) glial cells are involved in the onset of tolerance [19], we hypothesized and proved that RBL-2H3 mediators activate astrocytes, while PEA counteracts this activation by down-modulating RBL-2H3 degranulation. 

Mast cells are immune cells increasingly recognized to play pivotal roles not only in allergy, but also in host defense, innate and acquired immunity, homeostatic responses, [51] and inflammation [52]. Mast cells are located in almost all tissues, particularly around blood vessels and in proximity of nerve endings [52], with the same distribution being observed at meningeal level, both in the spinal cord and brain [53,54]. The critical role of MCs in the generation and maintenance of neuropathic pain has recently been investigated [55]. In particular, MC degranulated mediators, including nerve growth factor [56], have been shown to activate and sensitize nociceptors [57,58] as well as stimulate particular pain pathways [59]. 

Here we used a validated model of rat MCs, i.e., the RBL-2H3 cell line, and confirmed that sensitization with anti-dinitrophenyl (DNP)-IgE is needed to induce a tunable degranulation, as previously reported by other groups [47,60,61,62]. Although opioids, such as morphine, are known to activate MCs in vivo [61,62,63,64] and codeine was shown to induce human MC degranulation in vitro [65], the effect of morphine on the RBL-2H3 cell line was not investigated yet. Here we showed that morphine was able to significantly and dose-dependently increase degranulation of DNP-BSA-primed RBL-2H3 cells, without exerting any cytotoxic effect at the 0.3–300 μM dose range. Interestingly, morphine-induced degranulation was not only monitored by the release of a well-known MC mediator, i.e., β-hexosaminidase [66], but also by immunofluorescence for tryptase, a serine protease stored in MC granules [67] and released upon MC activation [52,68]. Although the presence of tryptase in RBL-2H3 cells has been questioned [69], our finding is in line with previous studies confirming tryptase release in RBL-2H3 after appropriate stimulation [47,70]. 

Moreover, here we showed for the first time that RBL-2H3 cells express the µ-receptor. This is consistent with what has been previously shown in human MCs (both primary and cell line LAD2) [65]. Although the exact involvement of µ-receptors in morphine-induced MC degranulation was out of the scope of the present study and was not further investigated, it could be assumed that morphine effect depended on µ-receptors, even if alternative pathways were identified (e.g., Mas-related G protein-coupled receptor) [71]. 

In the present study, the ability of ultramicronized PEA to down-modulate MC degranulation, in response to DNP-BSA challenge, either alone or associated with morphine, was confirmed. In particular, the effect of ultramicronized PEA was monitored by β-hexosaminidase assay, tryptase immunofluorescence, histamine quantification and RT-PCR for newly synthesized mediators (i.e., CCL2 and TNF-α). 

Within pre-formed granules, histamine is one of the main molecules produced by MCs [72], acting as a pro-inflammatory factor [73]. Our results demonstrated a time-dependent histamine release in RBL-2H3 cells, counteracted by PEA treatment up to 24 h of DNP-BSA stimulation. The result parallels with that obtained by another group in which histamine was measured by ELISA assay [50]. Moreover, the ability of ultramicronized PEA to decrease histamine levels was also observed by our group in a rat model of chronic neuropathic pain [49].

Notably, TNF-α may activate MCs as well as microglia and astrocytes, being involved in neuroinflammation and chronic pain [74], while CCL2 regulates recruitment of inflammatory cells [75] and acts as a chemoattractant to MCs [76]. The present results concerning the effect of PEA on RBL-2H3 are consistent with previous findings showing that this lipid amide negatively controls releasability of activated MCs, both in vitro [50,77,78], ex vivo [79] and in vivo [80,81], in accordance with the so-called ALIA mechanism originally postulated by the Nobel laureate Rita Levi Montalcini [82]. 

MCs can interact with virtually all cell populations in the CNS, particularly glial cells [53]. Interestingly, several lines of evidence indicate that chronic morphine treatment may directly or indirectly activate glial cells [13,19,83], resulting in the release of pro-inflammatory cytokines (such as TNF-α, IL-1β, IL-6 and chemokines), which in turn further sensitize the NMDA receptors, promoting the onset of morphine tolerance [84]. In particular, astrocytes are emerging as relevant players in the development of opioid tolerance [19,85,86] and morphine repeated treatment was found to significantly activate astrocytes in the spinal horn [87]. Consistent with previous findings [88,89], we here showed that primary rat cortical astrocytes express the μ-receptor and release calcium in response to morphine, thus confirming previous findings on the astrocyte-activating effect of morphine [90]. Notably, the lack of morphine toxicity towards astrocytes was also shown. 

The existence of a crosstalk between MCs and astrocyte in opioid tolerance development is an intriguing hypothesis that has not yet been evaluated. In the CNS, MCs are located at the meningeal [54] and blood brain barrier levels, in apposition to astrocytes and neurons and may enter the parenchyma when barrier permeability is increased [53,91]. Moreover, the powerful mediators released by MCs in the periphery can signal to the CNS compartment including astrocytes [92]. To study this relationship, morphine-activated cortical astrocytes were incubated with the conditioned medium of control or PEA pre-treated MCs. First, it was confirmed that MC medium did not alter astrocyte viability. Untreated MC medium led to a significant increase in gene expression of IL-1β, IL-6, CCL2, Serpina3n, EAAT2 and GFAP, in control and even more so in morphine-treated astrocytes, thus confirming the involvement of these non-neuronal cells in neuroinflammation [84]. Interestingly, the increase in the expression of astrocyte neuroinflammatory (but not proliferative, i.e., GFAP) genes was effectively counteracted by pre-treatment of MCs with ultramicronized PEA.

IL-1β is one of the first cytokines that have been involved in the onset of morphine tolerance: selective blockade of the signaling associated with IL-1β prevented the development of tolerance following chronic administration of morphine [93]. Moreover, once tolerance was established, intrathecal administration of an IL-1 receptor antagonist reversed hyperalgesia and prevented the additional development of tolerance and allodynia [53]. In addition, IL-1β knockout mice showed an increased morphine analgesia and prevention of morphine tolerance [93]. Accordingly, the decrease of IL-1β gene expression in response to incubation of morphine-treated astrocytes with PEA-treated MC medium might be considered as a protective effect of PEA against opioid-associated neuroinflammation. 

IL-6, whose production is induced by IL-1β and TNF-α, is also involved in morphine tolerance, as well as TNF-α itself, whose inhibition suppress the development of morphine tolerance in rats [94]. High levels of spinal IL-6 have been shown in tolerant rats [95] and repeated intrathecal administration of melanocortin 4 receptor antagonist reduced IL-6 expression and decreased astrocyte activation in morphine tolerant rats, counteracting the loss of morphine analgesic effect [96]. The decreased astrocytic gene expression of IL-6 in response to PEA-treated MC medium might again confirm the protective effect of PEA against neuroinflammation. As far as the chemokine CCL2 concerns, similar considerations could be drawn. Indeed, CCL2 is up-regulated during neuropathic pain conditions, while its neutralization (through intrathecal antibodies) improved morphine-induced analgesia [97]. 

Beside inflammatory cytokines and chemokines, other markers of astrocyte activation were analyzed here, to evaluate astrocyte phenotype and functions. SERPINA3 is an acute phase protein [98] belonging to the serpin superfamily of serine protease inhibitors [99]. SERPINA3 is synthesized mainly in the liver, lungs, and brain [100]. Here, activated astrocytes represent the main producers of SERPINA3 [101], whose expression is increased in response to IL-1 and TNF-α [102]. Previous experiments have shown that mice lacking serpina3n gene develop a more severe mechanical allodynia compared to wild-type mice [103]. Interestingly, Serpina3n was found to be upregulated in postmortem mid brain tissues from chronic cocaine abusers [104], and the astrogliosis induced by ischemic stroke-associated neuroinflammation was characterized by increased Serpina3n mRNA expression [105]. Furthermore, Serpina3n is involved in neuropathic pain [103] and was actually found to be upregulated in dorsal root ganglia after peripheral nerve injury [106]. With this in mind, the down-regulation of Serpina3n expression in morphine-treated astrocytes following incubation with the medium of PEA-treated MCs can be considered a further proof of the ability of ultramicronized PEA to rebalance the astrocyte-MC crosstalk. A possible explanation for this finding could be that ultramicronized PEA downmodulated TNF-α release from MCs, with this cytokine being responsible for the onset of morphine tolerance in vivo [94]. The hypothesis is supported by the in vivo evidence that TNF-α levels are lower in rats co-treated with morphine and PEA compared to those treated with morphine only [18].

Excitatory amino acid transporter 2 (EAAT2) is primarily localized on astrocytes and responsible for up to 90% of glutamate uptake in the brain [107]. Some evidence correlates a down-regulation of EAAT2 to chronic morphine treatment and pain-related conditions, although a defined role of this transporter is still unclear. In our experiments, EAAT2 over-expression on astrocytes may be viewed as an attempt to enhance glutamate up-take during neuroinflammation. 

Finally, the up-regulation of GFAP, i.e., the main intermediate filament protein of the astrocyte cytoskeletal compartment [108], is a key feature of astrocyte activation [108] responsible for both pathogenetic and protective roles [109,110]. While in vivo studies have mainly shown GFAP up-regulation in response to morphine chronic treatment [18,19,111], in vitro findings have been more inconsistent. In the present study, an up-regulation of GFAP in morphine-treated astrocytes was observed, with MC conditioned medium further increasing its expression. Surprisingly, the incubation of astrocytes with the conditioned medium from PEA-treated MCs resulted in an opposite effect on the expression of GFAP (i.e., increase) compared to all the other investigated genes, which indeed were decreased. In the light of the observed down-regulation of inflammatory genes, it might be inferred that in the evaluated in vitro system, GFAP increase in astrocytes is part of the PEA-mediated protective response to morphine potential damage.

## 5. Conclusions

In conclusion, the present study showed that morphine-induced MC degranulation leads to astrocyte activation, with MC mediators inducing astrocyte over-expression of genes involved in pain and inflammation. Pre-treatment with ultramicronized PEA decreased MC degranulation and reduced the expression of these astrocyte genes accordingly. Given the known involvement of astrocytes in the development and maintenance of morphine tolerance, our findings highlight one of the pathways through which PEA may control this phenomenon, balancing the crosstalk between MCs and astrocytes. This research lays the foundations for a more in-depth study on the molecular mechanisms sustaining PEA-induced MC down-regulation and modulation of glial cell behavior. 

## Data Availability

The data presented in this study are available on request from the corresponding author.

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
