# Peer review of "Ultramicronized N-Palmitoylethanolamine Regulates Mast Cell-Astrocyte Crosstalk: A New Potential Mechanism Underlying the Inhibition of Morphine Tolerance"

_biomolecules, 2023, doi:10.3390/biom13020233_

Round 1
Reviewer 1 Report
The manuscript submitted by Toti et al, it’s a well written paper. The authors demonstrated the effects of morphine on MC cell line (RBL-2H3) degranulation. They also show that PEA avoid this degranulation. In addition, they determined the effects of the conditionate medium from RBL cells over the astrocyte activation induced by morphine.
I think some items would be clarified:
1.- Figure 2a.- The authors showed the expression of MOR in RBL-2H3 cell line, I think they must include appropriate controls of MOR expression to strengthen this very important result. The information about the source of sample should be included, also, the number of catalogues of each antibody used in this work must include.
2.- Please explain why the concentration of antiDNP-IgE used in all the experiments is not showed in the optimization protocol?
3.- Figure 2b.- The immunofluorescence images lack of scale of magnification and they have low resolution. Similarly, figure 4e has this inconvenience and the DAPI stain is not clear.
4.- Why the astrocyte viability was not evaluated in the presence of conditionate medium from RHL-2H3 treated with PEA?
5.- What would happen to a longer exposure time of astrocytes to conditioned medium? Why not probed 24h exposition of this medium? Is it reversible the changes observed in astrocytes treated with the conditioned medium?
6.- The changes observed in the mRNA levels of some proinflammatories cytokines, are also observed to protein levels? It will be interesting to evaluated it.
7.- I suggest to the authors add to discussion section how does PEA avoid the degranulation of MC and how does the conditioned medium decrease the expression of proinflammatory cytokines in astrocytes ?(possible molecular mechanisms).
Reviewer 2 Report
Dear authors,
The manuscript entitled "Ultramicronized N-palmitoylethanolamine regulates mast cell-astrocyte crosstalk: a new potential mechanism underlying the inhibition of morphine tolerance” report on potential mechanism by which N-palmitoylethanolamine (PEA) delays opioid tolerance might depend on the control of the crosstalk between mast cells (MCs).
It presents scientific relevance for the area of Medicine, Biology and Biochemistry area. After consulting www.sciencedirect.com; https://pubmed.ncbi.nlm.nih.gov/ and others data base, authors have publications related to subjects related to the theme of the manuscript. The language (English) are satisfactory (I suggest the final revision)! However, you need to change some details/information in the abstract, Introduction, Methods, results, discussion and conclusions.
1. Abstract: Adequate, but I suggest rewrite and add information:
- The abstract is well written, with details of the methods used. However, I suggest inserting the results (numeric values) obtained, more relevant.
- I suggest highlighting the "innovative" proposal of the study, as well as the advantages / disadvantages, at the end of the abstract.
2. Introduction section: It is well written, but I suggest:
- Pages 1-2, lines 30-70: Long paragraph! I suggest splitting!
- Page 2, lines 72-77: I suggest joining the paragraphs!
- I suggest highlighting the "innovative" proposal of the study, as well as the advantages / disadvantages, at the end of the introduction.
3. Material and Methods section: The methodological proposal is appropriate to the manuscript, but I suggest:
- Pages 3-6, from “2.4. Cell viability assay” to 2.13 section: I suggest indicating the references used for the procedures adopted.
- Page 3, in “2.4. Cell viability assay” section: The authors used the MTT for the assays. Is this methodology already standardized by the research group in the laboratory? The "AlamarBlue - Cell Viability Assays" method could be an option as well.
- Page 4: in “2.7. Histamine assay” section: The authors used HPLC-MS/MS. What are the optimized flow parameters during the analyses? What are the analytical validation parameters used? Has the proposed method been validated? If so, which protocol / guidelines did you follow? What are the validation parameters studied? Precision, accuracy, LOD, LOQ, robustness, etc. What concentration levels are used to assess accuracy? I suggest detailing the proposed method in more detail...
4. Results section (or “Results and discussion”)?
Wouldn't it be more interesting to combine the "results” with the "discussion" to better describe the findings and compare them with other works published in the literature?? I suggest expanding the discussions!
- Pages 7 and 9: I suggest improving the quality of Figure 1 (1a and 1b) and Figure 3.
5. Discussion section
The entire section is written in a single paragraph! This is exhausting for the reader! I suggest linking the “discussions with the results”, discussing them after the figures.
- In results (Page 9): The authors write in lines 352-354“...Figure 4a, histamine concentration (µM) in treated and control cells was respectively 2.55 ± 0.1 vs 10.2 ± 0.13 at 30 minutes, 6.86 ± 0.71 vs 15.07 ± 0.5 at 2h, 24.57 ± 0.1 vs 39.03 ± 0.7 at 6h and finally 89.98 ± 0.9 vs 126.92 ± 0.9 at 24h”. However, no discussion about these data. I suggest expanding the discussions on the data obtained in Figure 4 with literature.
- I suggest, at the end of the "results and discussion", to write a paragraph summarizing the findings and their impacts on the research proposal.
6. Conclusion section
- Adequate, but I suggest highlighting the "innovative" proposal of the study, as well as the advantages / disadvantages/limitations.
* Figures: Adequate! Please, to improve the quality of figures.
* Supplementary data: Adequate!
* References: Please, check if the references are in accordance with the journal's rules.
Reviewer 3 Report
The authors collected interesting data demonstrating the effects of PEA on morphine tolerance. Timely paper providing new data useful to better design a therapy plan for people suffering from pain, with the aim to avoid opioid tolerance therefore addiction. Overall the experiments are well-designed, however I think minor changes are required to make the paper easy to read and understand to a broader audience.
Abstract: the abstract dives deep into the topic right away, a sentence about pain management will make it easier to get the purpose of the paper. Indeed, in the first section of the introduction the authors are spending a lot of time talking about pain.
Materials and methods:
- Line 85: I’m assuming you made a stock solution in DMSO, you should specify what’s the DMSO % in the final concentration you are using to treat the cells. I’m assuming is 1% from one og the graphs, but it should be stated in this section.
- Line 89: there is a verb in future while the rest of the section is in past tense
- Histamine assay section: the authors should state which brand is the instrument, the volume of injection, the mass transition they used to analyzed histamine, as well as limit of detection (LOD) and quantification (LOQ). Additionally, they should report the solvents with their names, not with the chemical formula (e.g., methyl cyanide instead of CH3CN).
Results:
- figure 2a: I highly recommend the authors to show the entire blot with a visible ladder.
- line 419: nuclei is misspelled
Discussion:
The authors should discuss their data to highlight their translational value. Therefore, they should discuss how the PEA dose range would translate in an in vivo model, and how their results contribute in the big picture of the topic they are presenting.
Round 2
Reviewer 1 Report
Thanks for attending all my suggestion.